# Randomized Double-Blind Controlled Trial on the Effect of Proteins with Different Tryptophan/Large Neutral Amino Acid Ratios on Sleep in Adolescents: The PROTMORPHEUS Study

**DOI:** 10.3390/nu12061885

**Published:** 2020-06-24

**Authors:** Oussama Saidi, Emmanuelle Rochette, Éric Doré, Freddy Maso, Julien Raoux, Fabien Andrieux, Maria Livia Fantini, Etienne Merlin, Bruno Pereira, Stéphane Walrand, Pascale Duché

**Affiliations:** 1Clermont Auvergne University, Laboratory of Adaptations to Exercise under Physiological and Pathological Conditions (AME2P), 63000 Clermont-Ferrand, France; Oussama.SAIDI@etu.uca.fr (O.S.); Eric.Dore@uca.fr (E.D.); 2Center for Research in Human Nutrition Auvergne, 63000 Clermont-Ferrand, France; 3Department of Pediatrics, Clermont-Ferrand University Hospital, 63000 Clermont-Ferrand, France; e_rochette@chu-clermontferrand.fr (E.R.); e_merlin@chu-clermontferrand.fr (E.M.); 4Clermont Auvergne University, INSERM, CIC 1405, CRECHE unit, 63000 Clermont-Ferrand, France; 5Toulon University, Laboratory of the Impact of Physical Activity on Health (IAPS), 83000 Toulon, France; 6Rugby Training Center of the Sportive Association Montferrandaise, 63000 Clermont-Ferrand, France; fmaso@asm-omnisports.com; 7OXSITIS LAB-NUTRITION, Chrono-Nutrition Food Supplements, 63110 Clermont-Ferrand, France; julien.raoux@oxsitis.com (J.R.); fabien.andrieux@oxsitis.com (F.A.); 8Neurophysiology Unit, Neurology Department, Clermont-Ferrand University Hospital, 63000 Clermont-Ferrand, France; mfantini@chu-clermontferrand.fr; 9NPsy-Sydo (EA 7280), Clermont Auvergne University, 63000 Clermont-Ferrand, France; 10Clermont Auvergne University, INRA, UMR 1019 UNH, ECREIN, 63000 Clermont-Ferrand, France; 11Biostatistics Unit (DRCI), Clermont-Ferrand University Hospital, 63000 Clermont-Ferrand, France; bpereira@chu-clermontferrand.fr; 12Clermont Auvergne University, Clermont-Ferrand University Hospital, INRAE, UNH, F-63000 Clermont–Ferrand, France; stephane.walrand@inra.fr

**Keywords:** nutrition, DLMO, metabolism, slow wave sleep, youth, pediatrics

## Abstract

Disturbed sleep is common in adolescents. Ingested nutrients help regulate the internal clock and influence sleep quality. The purpose of this clinical trial is to assess the effect of protein tryptophan (Trp)/large neutral amino acids (LNAAs) ratio on sleep and circadian rhythm. Ingested Trp is involved in the regulation of the sleep/wake cycle and improvement of sleep quality. Since Trp transport through the blood–brain barrier is competing with LNAAs, protein with higher Trp/LNAAs were expected to increase sleep efficiency. This randomized double-blind controlled trial will enroll two samples of male adolescents predisposed to sleep disturbances: elite rugby players (n = 24) and youths with obesity (n = 24). They will take part randomly in three sessions each held over a week. They will undergo a washout period, when dietary intake will be calibrated (three days), followed by an intervention period (three days), when their diet will be supplemented with three proteins with different Trp/LNAAs ratios. Physical, cognitive, dietary intake, appetite, and sleepiness evaluations will be made on the last day of each session. The primary outcome is sleep efficiency measured through in-home electroencephalogram recordings. Secondary outcomes include sleep staging, circadian phase, and sleep-, food intake-, metabolism-, and inflammation-related biochemical markers. A fuller understanding of the effect of protein Trp/LNAAs ratio on sleep could help in developing nutritional strategies addressing sleep disturbances.

## 1. Introduction

Adolescents are vulnerable to poor sleep [1,2]. A high social rhythm disruption, favorable during adolescence, would be associated to a substantial desynchronization in one’s routine that would likely leads to erratic sleep–wake pattern or promote sleep disruption among this population [3]. Obesity and elite sport are two factors that have been separately associated with sleep disturbances. They have a negative impact on holistic development, with lowered performance, both physical (recovery, metabolism, growth, and weight control) and cognitive (learning, memory, decision-making, and vigilance). Several studies show that obesity is associated with decreased sleep efficiency and increased arousal [4,5]. This impairs energy balance control by increasing energy intake through a modified sense of satiety and hunger [6]. In parallel, recent studies report that sleep disturbances are widespread in young athletes especially those practicing contact sports such as rugby [7,8]. Poor sleep impairs their athletic performances [9] and substantially increases the risk of injury, burnout, and concussion [10,11].

Besides endogenous factors linked to maturational changes during adolescence which push for a delay in the timing of sleep and increase the risk of sleep disturbances [1]. Other external factors specific for each population (obese or athlete) exacerbate poor sleep outcomes. In adolescents with obesity, the involvement of a complex interaction among sleep, obesity, metabolic disruption, and low grade inflammation was advanced [12,13]. On the other hand, the increased time demands by elite sports pressures including high training demands, evening intense exercise, competition, travel, and strict schedules contribute to excessive stress, anxiety, physiological excitement (e.g., increased heart rate at bedtime), pain, and muscle soreness which impair sleep in young athletes [7,14].

To date, few non-pharmacological approaches have been implemented to address sleep disturbances in these populations. A wide range of papers emphasize that external Zeitgeber such as nutrients and physical activity play a part in regulating the internal clock and determining sleep quality [15,16]. Nutrients reprogram peripheral clocks such as the liver or the gut clock by exerting an effect on the transcriptional and translational regulation of molecular clocks (e.g., CLOCK/BMAL) which generates a considerable reorganization of homeostasis and specific metabolic pathways [17]. Moreover, we have recently noted a growing interest in the effect of dietary intake on sleep and several reviews have been published over the last decade [18,19,20,21,22]

One of the macronutrients whose effect on sleep remains controversial is protein. Some amino acids, besides their use as substrates for protein synthesis, also act as precursors for the biosynthesis of compounds involved in various physiological processes. Experimentally, the availability of amino acid precursors can influence the rate of neurotransmitter biosynthesis [23,24,25,26]. However, the functional consequences of the intake of amino acid precursors on the nervous system are still imperfectly known. One study reported that ingested tryptophan (Trp) is involved in the regulation of the sleep–wake cycle and the improvement of sleep duration and quality [27]. Others have found that high protein intake, especially in the evening, is associated with increased alertness. This has been explained by the stimulating effect of certain amino acids on excitatory hormones such as adrenaline and noradrenaline [28,29,30]. Although it is hard to draw definite conclusions, it has also been suggested that Trp transport through the blood–brain barrier (BBB) competes with large neutral amino acids (LNAAs). This could explain why high protein intake decreases Trp bioavailability in the brain and increases alertness [31]. Previous studies on tryptophan depletion have shown that decreasing circulating tryptophan impairs sleep (e.g., increased arousal and delayed REM sleep) [32,33]. Moreover, it was suggested that small amount of dietary Trp intake could improve sleep efficiency in insomniac subjects [34]. The pilot study by Ong et al. (2017) also reported favorable effect of Trp rich protein on accelerometry measured sleep among young adults without sleep complaints [35].

Recent reviews strongly recommended nutrition as a basic option to improve sleep, especially among the population with sleep disturbances [36,37]. A more comprehensive understanding of the relationship between dietary protein (Trp/LNAAs ratio) and sleep could help in developing nutritional strategies to address adverse effects on adolescent sleep, especially among vulnerable persons such as adolescents with obesity or junior elite athletes. Here, we describe the design of the PROTMORPHEUS study and highlight the importance of carrying out this trial under free-living ecological conditions.

## 2. Materials and Methods

### 2.1. Study Participants, Eligibility Criteria and Recruitment Procedure

Adolescents will be recruited and evaluated throughout the study at three sites: The Department of Pediatrics, University Hospital, Clermont-Ferrand, France; the Montferrand Sports Association—Rugby Section, Clermont-Ferrand, France; and the Pediatric Obesity Center (Tza Nou, La Bourboule, France). Male adolescents aged 14–17 years over Tanner Stage 3 will be enrolled in this study who are rugby players engaged in the under 17 years national category or with obesity (BMI) ≥ 95th percentile. A total of 48 adolescents (24 rugby players and 24 with obesity) will be recruited. Eligibility criteria are presented in Table 1. A physician will assess eligibility in the Department of Pediatrics, University Hospital, Clermont-Ferrand, France. All diagnosed sleep disorders will be considered as exclusion criteria and all recruited adolescents will be screened for Obstructive Sleep Apnea using the Berlin questionnaire [38]. Participants will also be screened for depression using the Kutcher Adolescent Depression Scale (KADS) where the score ≥ 6 will be considered as exclusion criteria [39]. Moreover, they will not present any eating disorder according to the Dutch Eating Behavior Questionnaire (DEBQ) [40].

This study will be based at both “Tza Nou” (Obesity Treatment Center for Children and Adolescents) where recruited adolescents with obesity are housed, and the Rugby Training Center of the Montferrand Sports Association for young elite rugby players. The investigators will travel to the sites with the laboratory equipment for ambulatory running of the study. The adolescents will be informed and agree to take part in the study with parental written consent. The experimental design complies with the ethical principles in the 2008 revision of the Declaration of Helsinki and has been approved by an ethics committee (comité de protection des personnes (CPP) Sud Meditérrannée II, registration number: 2018 A23) prior to the study launch (clinical trial No. NCT04041934).

### 2.2. Randomization

Treatments (Trp/LNAAs ratio: PROT-REF = 0.04, PROT1 = 0.07, PROT2 = 0.11) will be assigned by randomization at a ratio of 1:1:1 in each sample. Stratification will be performed every six subjects (one experimental group). Each participant will have a single identification number will identify each participant. The participant number will be assigned chronologically on the randomization list. A comprehensive document describing the randomization procedure will be safeguarded at the Children’s Clinical Research Centre, INSERM CIC 1405. Indistinguishable protein containing different Trp/LNAAs ratio will be sent by Oxsitis (Clermont-Ferrand, France). Investigators, patients, and evaluator will be blinded to the patient’s treatment allocation. A computer program will generate the coding list and will allocate coding numbers to participants from the specific trial site. The data will be collected and recorded on case report forms (CRFs) by local research coordinators blinded to the randomized intervention. A trained research coordinator, also blinded to the randomized intervention, will centralize and record data in an electronic database.

### 2.3. Design Overview

PROTMORPHEUS is a randomized, double-blind, controlled, monocentric trial in which each subject acts as his own control to reduce interindividual variability.

#### 2.3.1. Objectives

The primary objective of this study is to investigate the effect of proteins with different Trp/LNAAs ratios on sleep efficiency (Figure 1). Secondary objectives are to determine the effect of protein Trp/LNAAs ratio on:Other sleep outcomes such as sleep staging, sleep onset latency (SOL) and wake after sleep onset (WASO)Circadian phase measured by DLMOMorning cortisol peakBiochemical markers related to sleep, food intake, metabolism, and inflammationMetabolic fitness measured by indirect calorimetry on submaximal testMuscle fatigue measured by time of maintenance of muscle strengthCognitive performances measured by a battery of tests including Stroop, trail marking, barrage, California verbal learning, and multiple object trackingEnergy intake and proportion of that energy derived from each macronutrient class (carbohydrate, fat, and protein) measured on meals offered ad libitumAppetite sensations (visual analog scales)Sleepiness rate (Karolinska scale)

#### 2.3.2. Experimental Sessions

At baseline, participants will undergo numerous assessments for anthropometry, body composition, and resting metabolic rate (RMR). They will complete the Morningness–Eveningness Questionnaire for Children and Adolescents (MEQ-CA) to determine their chronotype [41], the self-report Pittsburgh sleep quality index (PSQI) to measure subjective sleep quality [42], the Epworth sleepiness scale (ESS) to measure daytime sleepiness [43], and the international physical activity questionnaire (IPAQ-SF) to estimate habitual physical activity level [44].

After the baseline and one week before to the experimental session launch of each group, time in bed (TIB) will be fixed, and sleep–wake cycle will be continuously monitored using accelerometry. Participants will also be equipped with the Sleep Profiler-PSG2 TM (Advanced Brain Monitoring, Carlsbad, CA) to familiarize them with sleeping with the device. Thereafter, three sessions will be conducted. Each session will be held over a week (Figure 2). Given the potential effect of electronic media (smartphone, laptop, etc.) on sleep and circadian rhythm [45], their evening use will be prohibited during the study.

To limit inter-individual differences, participants will undergo a washout period (three days). During this period, their diet will be set to the recommended dietary allowance (RDA) (1.8 × basal metabolism for athletes and 1.3 × basal metabolism for adolescents with obesity) with protein intake fixed at 1.2 g.kg^−1^. The RDA will be calculated individually for each subject. Three meals and a snack will be offered each day. The meal trays will be prepared in advance and will be the same between sessions. At the last night of washout (Night 3), five saliva samples will be collected to capture DLMO (3, 2, and 1 h before bedtime, bedtime, and 1 h after bedtime). Participants will be fitted with the Sleep Profiler-PSG2 TM for the night. On awakening next day, two saliva samples (on awakening and 60 min later) will be taken to measure morning cortisol level, and blood samples will be collected in the fasting state to measure several biochemical markers related to sleep, food intake, metabolism, and inflammation. During the next three days, dietary intake will be supplemented with protein shakes (PROT-REF = 0.04, PROT1 = 0.07, or PROT2 = 0.11). From Day 4 to Day 6, the diet will be iso-energetic to the washout period (Days 1—3) diets. However, protein intake will be raised from 1.2 to 1.6 g.kg^−1^ through (PROT-REF, PROT1, or PROT2) shakes and intake from other macronutrients will be adjusted accordingly. Similar to Night 3, participants will collect five saliva samples and undergo electroencephalogram (EEG) recordings during Night 6.

At Day 7 (Figure 3), on awakening, saliva samples and blood samples will be collected. Food will then be offered ad libitum in four meals (breakfast, lunch, afternoon snack, and dinner) representing 24 h energy intake. Participants will be asked to fill out visual analog scales for their appetite sensations and sleepiness level throughout the day. After breakfast, participants will take part in evaluations including metabolic fitness on submaximal test, muscle fatigue measured by time of maintenance of strength of the knee extensor muscles, and cognitive performance measured by a battery of tests (Stroop, trail marking, trail-making task, barrage test, California verbal learning, and multiple object tracking).

Protein intakes during the first three days of each experimental session will be 1.2 and 1.6 g.kg^−1^ the following three days. The average nutritional requirement (RDA) for tryptophan is 4 mg.kg^−1^.day^−1^. The French food safety agency’s opinion on the use of tryptophan in dietary supplement indicates a threshold of 200 mg.day^−1^. The protein supplementation used in this study will not exceed this tryptophan threshold.

### 2.4. Measurements

#### 2.4.1. Baseline Assessments

Anthropometric characteristics and body composition

Body mass (BM) will be measured with a digital scale, standing with minimal clothing. Height will be measured barefoot using a stadiometer. Body mass index (BMI = body mass divided by height squared in kg.m^-2^) will then be calculated. Fat mass, visceral fat mass, and fat-free mass will then be established using dual-energy X-ray absorptiometry (DXA) scans (QDR4500Ascanner, Hologic, Waltham, MA, USA).

Resting metabolic rate

Resting metabolic rate (RMR) will be measured in the fasting state, using indirect calorimetry (MetaMax 3b, Cortex Biophysik, Leipzig, Germany). As recommended by the manufacturer, gas analysis will be calibrated before the test. The test will be held in a thermoneutral environment (22–25 °C). Participants will lie in a supine position for 45 min before starting the measurements. After achieving a steady state, O_2_ consumption and CO_2_ production standardized for temperature, barometric pressure, and humidity will be recorded at 1-min intervals for 20–45 min and averaged over the whole measurement period. RMR (kcal/day) and respiratory quotient (CO_2_/O_2_) will then be calculated.

Morningness–Eveningness Questionnaire for Children and Adolescents (MEQ-CA)

Participants will be asked to complete an MEQ-CA which is a 19-item questionnaire to assess habitual waking and bedtimes together with the times of day when they prefer to “perform” [41]. The MEQ-CA score ranges from 16 to 86, with scores below 42 identifying participants as evening types, scores between 42 and 58 as intermediate types, and scores above 58 as morning types.

Pittsburgh sleep quality index (PSQI)

The PSQI will be used to measure sleep quality [42]. It is a 19-item self-report instrument designed to measure sleep quality and disturbance over the previous four weeks. The score range of the questionnaire is 0–21; higher scores (>5) indicate higher levels of sleep disturbances.

Epworth sleepiness scale (ESS)

EES is a short, self-administered questionnaire commonly used to quantify overall excessive daytime sleepiness and evaluate tendency to fall asleep in eight sedentary situations [43]. The total daytime sleepiness score ranges from 0 to 24 (0–7 normal, 8–9 mild excessive, 10–15 moderate excessive, and 16 or above severe excessive).

International physical activity questionnaire, short form (IPAQ-SF)

The short interviewer-administered IPAQ contains seven questions that measure the frequency (days/week) and duration (minutes/day) of vigorous and moderate-intensity activities and walking in bouts of at least 10 min in the last week [44]. Physical activity outcomes from the IPAQ-SF for this study were as follows: weekly minutes spent in vigorous physical activity, moderate-intensity physical activity, walking, and sedentary time.

#### 2.4.2. Outcome Measures

Sleep

The Sleep Profiler-PSG2™ (Advanced Brain Monitoring, Carlsbad, CA, USA) will be used to measure sleep [46]. This minimally invasive device is approved by the Food and Drug Administration (FDA). Its measure is reproducible and validated for gold standard laboratory polysomnography (PSG) in adults and children aged > 6 years. It has three channels: electroencephalography, electro-oculography, and electromyography from frontopolar sites; airflow through a nasal cannula and pressure transducer; head movement and position by actigraphy; snoring with an acoustic microphone; pulse from the forehead and finger; wireless wrist oximetry; and thorax and abdomen effort by respiratory inductive plethysmography. The participants will be equipped by investigators. The records will then be extracted through the Sleep Profiler portal. Validated auto-staging will be applied based on the ratios of the power spectral densities and auto-detection of cortical and microarousals, sleep spindles and ocular activity [47,48]. After data processing, an experimented sleep expert will review the recordings to confirm the accuracy of the auto-sleep staging. The Sleep Profiler-PSG2™ provides total sleep time (TST), sleep onset latency (SOL), wake after sleep onset (WASO), sleep efficiency (SE), number of awakenings lasting more than 30 s, arousal index, and time spent in non-rapid (NREM; Stage-1, Stage-2, and Stage-3) and rapid eye movement (REM) according to American Academy of Sleep Medicine (AASM) recommendations.

Circadian phase and morning cortisol

Five salivary samples will be collected in the participant’s habitual sleep environment in dim light conditions (<20 lux in any direction of gaze) according to the method described by Mandrell et al. (2018) [49]. Participants will allow saliva to collect in their mouth and then drool into a collector tube rather than spitting (Salivette^®^ Sarstedt). Evening saliva samples will be taken for the assessment of dim light melatonin onset (DLMO), a measure of circadian phase. To record the increase in melatonin, participants will be asked to start collecting every hour (3, 2, and 1 h) before bedtime. Saliva will be collected again at bedtime and investigators will wake the participants to take a final sample 1 h after bedtime. The following morning two saliva samples will be taken (on awakening and 1 h after awakening) to measure cortisol peak. The saliva collection tubes will be coded according to the biomarker and collection time (melatonin 3, 2, and 1 h before bedtime, melatonin at bedtime, melatonin 1 h after bedtime, cortisol on awakening, and cortisol 1 h after awakening). After each hourly collection, the tube will be placed in a labeled box in a freezer. The samples will then be stored in a freezer at −80 °C until analysis and treatment by expanded-range enzyme immunoassay.

Energy expenditure and physical activity

Physical activity will be monitored during the intervention using ActiGraph GT3X accelerometers (Pensacola, FL, USA) at a 1-s epoch setting to ensure that participants keep the same level of physical activity between sessions. This multisensory tool continuously collects various motion parameters through a triaxial accelerometer [50]. Sex, age, body weight, and height data will be transferred to estimate the level of physical activity and activity energy expenditure using algorithms developed by the manufacturer’s software (Actilife). Accelerometers will be secured to belts and distributed to the participants on awakening on Day 1. They will be instructed to wear the instruments on their waist until Day 7, and told to remove the accelerometer only when there may be water contact (showering).

Biochemical measures

Venous blood samples will be taken via an indwelling catheter at 07:30 am in the fasting state. Samples will be collected in EDTA K2 Vacutainer^®^ tubes (Becton Dickinson, Franklin Lakes, NJ, USA), Vacutainer^®^ Lithium Heparin tubes (Becton Dickinson, Franklin Lakes, NJ), and in Vacutainer^®^ SST II Advance tubes (Becton Dickinson, Franklin Lakes, NJ, USA). Samples collected will be immediately centrifuged at 3000× *g* for 10 min at 4 °C except for blood samples on SST tubes, which will be centrifuged after 30 min. After centrifugation, the serum aliquots will be frozen and stored at −80 °C until analysis for cytokines; all other blood samples will be analyzed immediately.

Blood samples will be taken at two time points (Days 4 and 7) in each session for the analysis of glucose, cholesterol triglycerides, urea, serum glutamic oxaloacetic transaminase (AST), serum glutamic pyruvic transaminase (ALT), insulin, leptin, ghrelin, adiponectin, GLP-1, PYY-36, nesfatin, irisin, GH, GHRH, TSH, serotonin, kynurenine, CRP, IL-1β, Il-6, TNF-α, IL-4, IL-10, sTNFR1, and sTNFR2. Plasma amino acids will be analyzed. The plasma Trp/LNAAs ratio will be calculated as Trp concentration (in μmol/L) divided by the sum of LNAAs. Changes in all biomarkers will be measured as the differences between Days 4 and 7 of each session.

#### 2.4.3. Secondary Assessments

On awakening on Day 7, the participants will be managed by investigators to take part in the different evaluations in the same order at each session (Figure 3). They will be given appetite and sleepiness questionnaires to complete throughout the day. After saliva sample and blood sample collection, they will have an ad libitum breakfast. They will then be allocated to three workshops for metabolic fitness, muscle fatigue, and cognitive performance testing following the same order between sessions. Finally, their energy intake will be measured by ad libitum meals for lunch, afternoon snack, and dinner.

Metabolic fitness

After sitting quietly for 20 min, subjects will perform, to the point of volitional fatigue, a graded exercise test on an electromagnetically braked cycle ergometer with continuous gas collection and heart rate monitoring. After a 2-min warm-up involving unloaded pedaling, athletes will start at 60 W and obese participants at 40 W. The work rate will be stepped up by 15 W every 3 min. If the heart rate is unstable, this stage will be extended for up to 5 min to obtain a heart rate stable to ±5 beats. When the respiratory exchange ratio (RER) is greater than or equal to 1.00—indicating no fat oxidation—the work rate will be stepped up by the same increments at 1-min intervals until volitional fatigue is reached. The VO_2_ peak will be considered to have been reached when the RER is greater than or equal to 1.05, and the subject has achieved his age-predicted maximal heart rate (HRmax: 220—age), according to the methodology validated by Riddell et al. (2008) [51]. All the tests will be performed on a portable cycle ergometer (Monark 828E, Goteborg, Sweden). The O_2_ consumption (VO_2_) and CO_2_ production (VCO_2_) will be measured breath-by-breath through a mask connected to an O_2_ and CO_2_ analyzer (MetaMax 3b, Cortex Biophysik, Leipzig, Germany). Ventilatory parameters will be averaged every minute during the submaximal exercise test and the subsequent 10-min recovery period. Heart rate will be monitored continuously throughout the experiments (Polar RS800cx monitor, Polar, Finland).

Muscle fatigue

Subjects will be placed on a dynamometer equipped with a force sensor. The back and hip of the participants will be attached to the back of the chair with a belt to prevent compensatory movements that might bias the force measurements. The angle formed at the hip will be 40° (0° being the angle in the standing position). The participants’ right leg will be firmly attached above the malleolus, and the force sensor screwed to the leg clip. For the maximal voluntary isometric contraction (MVIC) test, participants will perform MVIC of knee extensors (duration = 3 s, recovery = 60 s) on the dynamometer at randomized angles of 66°, 76°, and 87° (0° = full extension, two tests at each angle) to determine the optimal angle. After a standardized 10-min warm-up period at the optimum angle, participants will then maintain a voluntary (static) isometric contraction of 60% of maximum strength. The exercise will end when subjects are unable to keep the required level of force longer than 3 s despite strong verbal encouragement [52,53]. Fatigue and recovery levels will be assessed immediately with an MVIC after exercise and then at 3, 6, and 15 min after cessation of exercise. Neuromuscular fatigue will be determined from the time of contraction retention. The difficulty of the effort will be assessed during the task using a child and adolescent’s subjective perception scale (CERT). The duration of this evaluation is estimated at 30 min.

Cognitive performance

A complete battery of tests including the Stroop color–word test, trail-making task (TMT), barrage test, California verbal learning test (CVLT), and multiple object tracking (NeuroTracker, CogniSens Athletic Inc., Montreal, Canada) will be used to measure cognitive performance (Table 2).

Ad libitum 24 h energy intake

Food will be offered ad libitum in four meals representing 24 h energy intake. The composition of the buffet meal will conform to the adolescents’ tastes as determined by the food questionnaire filled out at baseline. Top-rated items will be avoided to limit excessive intake that might overwhelm the effect of the trial on food intake. The buffets offered the participants will be the same between adolescents and sessions. Sufficient amounts of food and identical meals will be offered for the three sessions, and participants will be asked to eat until satiated; additional food will be available if desired. Food consumption will be weighed and recorded by investigators. Energy intake and macronutrient and micronutrient composition will then be calculated using a professional computerized nutrient analysis program (Bilnut 4.0 SCDA Nutrisoft software) and Ciqual tables (2017 version).

Subjective appetite sensations

Appetite sensations will be collected throughout the day using visual analog scales (150 mm scales). Adolescents will report their hunger, fullness, desire to eat, and prospective food consumption at 13 regulated times: on awakening; before and immediately after breakfast, lunch, snack, and dinner; and at bed time. The questions will be “How hungry do you feel?”, “How full do you feel?”, “Would you like to eat something?”, and “How much do you think you can eat?”. Adolescents will be asked to respond on a scale from “not at all” to “a lot”. This method has been previously validated [56].

Subjective diurnal sleepiness

The Karolinska sleepiness scale will be used to assess alertness/sleepiness at 13 regulated times throughout the day: on awakening; before and immediately after breakfast, lunch, afternoon snack, and dinner; and at bed time. A nine-point verbally anchored scale ranging from 1 to 9 (“extremely alert”–“extremely sleepy”–“fighting sleep”) will be used. Higher scores indicate higher sleepiness [57].

### 2.5. Data Management

Data will be handled in compliance with French law, and collected and recorded on CRFs. A trained research coordinator will record data in an electronic database. All original records will be archived at trial sites for 15 years.

### 2.6. Statistical Analysis

#### 2.6.1. Sample Size Power

The justification of sample size estimation is based on the comparison of mean differences in sleep efficiency (primary outcome) measured pre- and post-intervention, between the three experimental sessions. Few studies report sufficiently robust information on intra-individual variability. We calculated that, for a type I error at 0.017 (correction due to multiple comparisons), a statistical power equal to 90%, and a coefficient intra-individual correlation of 0.5, 22 subjects per adolescent population recruited, either with obesity or athletes, would be needed to demonstrate an effect size of the order of 0.8. This seems reasonable in view of the work reported in the literature [35]. To take into account lost to follow-up, it was finally agreed to include 24 participants per population.

#### 2.6.2. Preliminary Analysis

Statistical analyses will be conducted with Stata v15 (StataCorp, College Station, TX, USA). Two-sided p < 0.05 will be considered statistically significant. Participants will be described at baseline and compared between randomized groups in terms of compliance with eligibility criteria, demographic characteristics, clinical characteristics, and treatments. The comparability of groups at baseline will be assessed on the main participant characteristics and potential factors associated with the primary outcome. A possible difference between groups in any of these characteristics will be determined by both clinical and statistical considerations. The number of participants included and the inclusion curve will be presented by group.

#### 2.6.3. Analyses for Preliminary Endpoint

Intention-to-treat analysis will be considered for the primary analysis. To prevent attrition bias, imputation of missing data is planned. The statistical analysis plan also provides for an additional per-protocol analysis.

To study the effect of quality of protein intake on quality of sleep, the primary analysis will be performed using statistical tests for repeated correlated data. Mixed models to take into account between and within subject variability will be used, considering the following fixed effects: study of protein (PROT-REF, PROT1, and PROT2), point-time evaluation, period, order, and carry-over together with their interactions (including *protein × point-time evaluation*), with subject as a random-effect. Then, the group effect (adolescent rugby players vs. adolescents with obesity) will also be studied, and special attention will be paid to the interaction status *group × proteins*.

These analyses will be followed if necessary (p < 0.05) by a post-hoc test suitable for multiple comparisons such as the Tukey–Kramer test. Multivariable analysis should be used to complete the aforementioned analyses, to take into account possible confounding factors including age and BMI. Results will be expressed in terms of effect sizes and 95% confidence intervals.

#### 2.6.4. Analyses for Secondary Endpoints

Secondary analyses will determine the effect of sleep quality on subsequent food intake and physical and cognitive performance. Thus, all the secondary quantitative endpoints (TST; SOL; WASO; variation in sleep stages; DLMO measured by salivary assays; blood markers of sleep, inflammation, and food intake; resting energy expenditure; 24 h food intake for the next day; measurement of energy and macronutrient intake assessed by the weighing method during meals consumed ad libitum; metabolic fitness; muscle fatigability; and cognitive performance assessed by the cognitive test battery) will be studied as described for the primary endpoint. These analyses will be carried out for each of the study groups (adolescent rugby players vs. adolescents with obesity). In a second step, the two groups of subjects will be pooled; the aforementioned statistical approaches will be used with an additional fixed group effect and the associated interactions.

A sensitivity analysis will be performed to study the statistical nature of missing data and to determine the most appropriate approach for imputing missing data.

## 3. Discussion

We describe the rationale and design of a randomized controlled trial to evaluate the effect of protein Trp/LNAAs on sleep in adolescents (either elite rugby players or with obesity). Emerging literature warns about the precariousness of adolescents’ sleep, especially for those practicing elite sport or with obesity. Cumulated demands of being both high school students and junior athletes may contribute to poor sleep, especially as elite sport pressures such as high training volume, late evening exercise, travel, and competition are linked to sleep disturbances [7]. For the second population, studies underlined that adolescents with obesity were particularly prone to several psychopathologic conditions, including anxiety and chronic stress, which affect the hypothalamic–pituitary–adrenal axis and increase the arousal/sympathetic nervous systems activities [58]. Furthermore, the literature points to a complex interaction between sleep, obesity and metabolic disruption [12,13]. Excess adiposity may lead to dysregulation of the adipose secretory factors that contribute to the development of low-grade systemic inflammation marked by an over-release of several pro-inflammatory markers. The results of the HELENA study in adolescents show an association between disturbed sleep and systemic inflammation [59].

The PROTMORPHEUS protocol also seeks to clarify whether sleep variation through protein with higher Trp/LNAAs will be accompanied by improved in physical and cognitive performance, along with dietary intake, appetite, and sleepiness sensations.

The effect of Trp on sleep was first derived from the serotonergic hypothesis (Figure 4). Serotonin (5-HT) is synthesized from Trp circulating in the brain by a two-step procedure in raphe neurons. 5-HT, in turn, is a precursor of melatonin, and both help regulate of sleep-wake behavior [60]. However, Trp is an essential amino acid whose sole source is the degradation of dietary proteins. The passage of Trp to the brain is ensured through carriers at the BBB. However, carrier transport depends on other competitive amino acids (LNAAs leucine, isoleucine, valine, phenylalanine, and tyrosine). Increased brain uptake of Trp does not therefore depend only on Trp but rather on the blood Trp/LNAAs ratio [37]. Given that melatonin and 5-HT are synthesized from Trp, we can reasonably expect an increase in their levels in the brain depending on the Trp/LNAAs ratio of the administered protein. This hypothesis has been validated in animal models [61]. Moreover, Minet-Ringuet et al. (2004) also showed that protein intake restored altered sleep effectively, especially slow-wave sleep (SWS) depending on protein Trp/LNAAs ratio [62]. However, to our knowledge, no studies have been done to test this hypothesis in humans. The study by Markus et al., (2005) found that evening intake of protein rich in Trp increased the ratio of plasma tryptophan to the sum of LNAAs with an improvement in next-day alertness and cognitive performance [63]. However, objective sleep quality and staging were not measured in that study.

PROTMORPHEUS has several strong points. To our knowledge, it is the first protocol investigating the effects of such an intervention in humans under ecological free-living conditions. Each adolescent participating in the study will be followed for three weeks during which their food intake, sleep, and physical activity will be controlled and recorded continuously. To improve the feasibility of the study, the methodology is designed to be minimally intrusive. Instead of conducting the study in the laboratory, which could make it extremely intrusive, we will let participants spend their nights in their habitual sleeping environment. More importantly, previous studies found that spending the night in laboratory conditions could introduce bias in sleep assessment. Ambulatory measures will therefore be used in the usual sleep environment, including minimally invasive electroencephalogram recordings and a simplified circadian phase assessment method. Recruited adolescents will be living at boarding schools and the investigators will move on-site to ensure that the measurements are conducted in appropriate conditions. Given the density of measures and the time constraint during evaluation (Day 7), workshops will be set up. Participants will switch between the workshops in an assisted way with the same order between the sessions.

Latent factors related to both obesity and/or sleep may affect outcomes of studies investigating the effect of dietary intervention on sleep. For instance, we already showed in previous works that physical activity could affect sleep in adolescents with obesity [64]. Therefore, we underline that these factors should be systematically quantified and added as confounders. Laboratory PSG is considered as the gold standard for sleep assessment. The absence of this measure is perhaps the main limitation of this study. However, gold standard PSG is not a viable option in the context of this research because: (i) it is cumbersome (two measures per session); (ii) differences in sleep quality may result from changes in sleeping environment; and (iii) the protocol already makes strong demands on the adolescent’s personal life during the protocol (three consecutive weeks). The ambulatory wearable device to be used includes all the major aspects of PSG: three EEG channels, airflow, thorax, and abdomen belt and pulse oximetry. Its measurements have been validated and yield results comparable to polysomnography in assessing sleep staging and several other parameters. However, this study is also limited by a small sample size and does not include female subjects. Although it is interesting to look for gender effect, the control of menstrual cycle phase in adolescent girls would be complicated and incompatible with such a study design. Previous studies have shown that menstrual cycle phase impacts sleep, circadian rhythm, and food intake, which could lead to biased assessments [65]. Finally, the effect of such an intervention in the medium and long term remains to be determined and would be interesting. Motivating the adolescents to follow this study protocol for a longer period is challenging and may severely penalize the feasibility of the study. However, previous studies already showed that dietary manipulation affect sleep significantly even over a shorter period of time than that of this study [66,67].

## 4. Conclusions

Given the prevalence of sleep disturbances during adolescence, particularly in specific populations such as athletes and teenagers with obesity, a non-pharmacological approach based on nutritional quality (protein with Trp/LNAAs ratio) could significantly improve duration and quality of sleep. Improvement of sleep could lead to better cognitive and physical performance and better daytime functioning and control of food intake.

## Figures and Tables

**Figure 1 nutrients-12-01885-f001:**
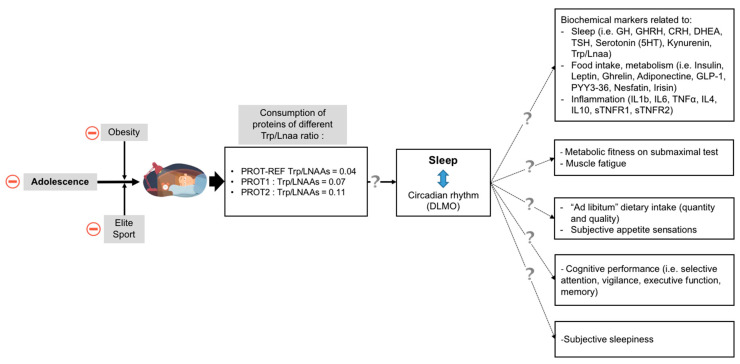
Study objectives (CRH, corticotropin-releasing hormone; DHEA, dehydroepiandrosterone; GH, growth hormone; GHRH, growth hormone releasing hormone; GLP-1, glucagon-like peptide 1; IL-1β, interleukin-1 beta; IL-4, interleukin-4; IL-6, interleukin-6; IL-10, interleukin-10; LNAAs, large neutral amino acids; PYY3-36, peptide YY-36; sTNFR1, soluble tumor necrosis factor receptor 1; sTNFR2, soluble tumor necrosis factor receptor 2; TNF-α, tumor necrosis factor alpha; Trp, tryptophan; TSH, thyroid-stimulating hormone).

**Figure 2 nutrients-12-01885-f002:**
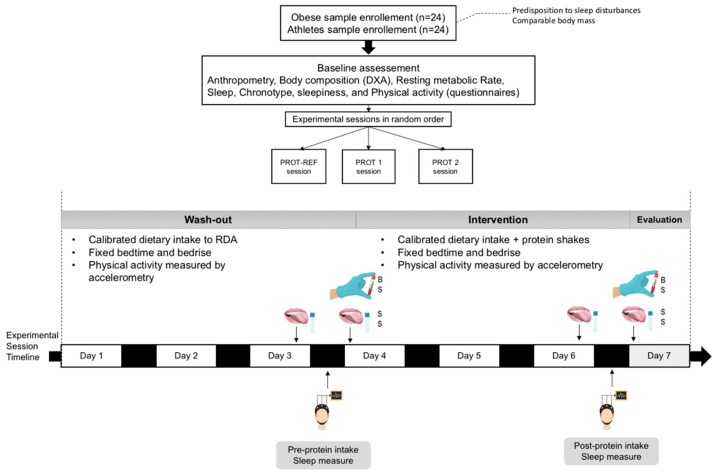
PROMORPHEUS protocol overview (BS, blood sample; DXA, dual-energy X-ray absorptiometry; RDA, recommended dietary allowance; SS, salivary sample).

**Figure 3 nutrients-12-01885-f003:**
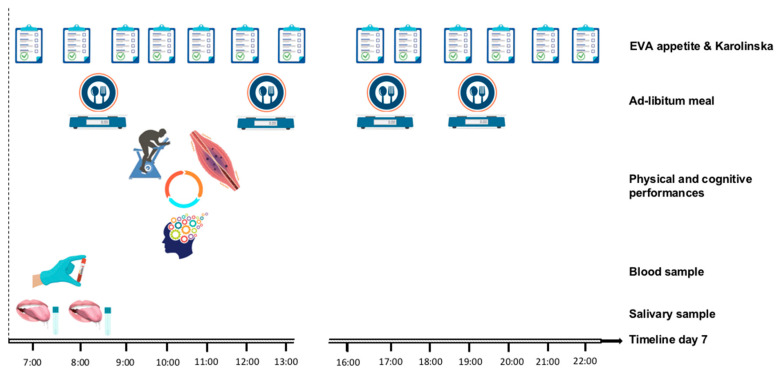
Timeline of Day 7 (evaluation).

**Figure 4 nutrients-12-01885-f004:**
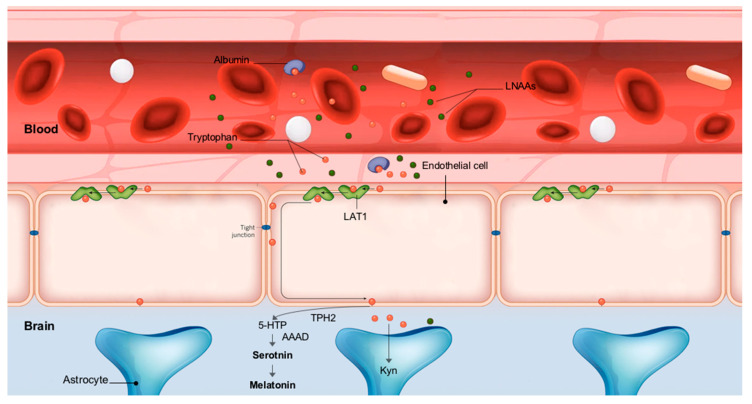
Tryptophan uptake and metabolism by the central nervous system: After degradation of dietary proteins, Trp is present in the bloodstream in two forms: either linked to albumin (50–80%) or in the free form. Only the free fraction of Trp can cross the BBB through active transporter LAT-1 present on the membrane of certain cell populations, mainly endothelial cells. However, Trp must compete with the other LNAAs when crossing the BBB. LNAAs can thus restrict tryptophan travel to the brain. The biosynthesis of 5-HT from Trp is carried out in two separate steps: The first is the hydroxylation of Trp to 5-hydroxytryptophan (5-HTP), a reaction catalyzed by the limiting enzyme of this biosynthetic pathway, tryptophan hydroxylase-2 (TPH2). Because tryptophan hydroxylase is typically 50% saturated with its tryptophan substrate, an increase or decrease in tryptophan availability in the brain can increase or decrease brain serotonin synthesis. This step is followed by decarboxylation of 5-HTP into 5-HT by L-aromatic amino acid decarboxylase (AAAD). In the pineal gland, this biosynthetic pathway is followed by the synthesis of melatonin from a fraction of the 5-HT produced, However, the main fraction of brain Trp is metabolized through the kynurenine pathway by microglia and astrocytes leading to the synthesis of kynurenic acid or quinolinic acid. (5-HTP, 5-hydroxy-tryptophan; AAAD, L-aromatic amino acid decarboxylase; BBB, blood–brain barrier; Kyn, kynurenin; LAT1, large neutral amino acid transporter 1; LNAAs, large neutral amino acids; TPH2, tryptophan hydroxylase-2).

**Table 1 nutrients-12-01885-t001:** Eligibility criteria.

**Inclusion Criteria**
Male adolescents aged 14–17 years Tanner stages 3–5 Athletes playing high-performance rugby or adolescents with obesity (BMI ≥ 95th percentile)
**Exclusion Criteria**
High risk of Obstructive Sleep Apnea (i.e., Berlin questionnaire with two or more categories where the score is positive) Diagnosed major sleep disorders (e.g., narcolepsy, children obstructive sleep apnea with apnea-hypopnea Index > 5, restless leg syndrome, REM sleep behavior disorder, Bruxism) Medical history inconsistent with the study (e.g., depression, attention-deficit/hyperactivity disorder, post-traumatic stress disorder, concussion) Chronic illness or injury that might interfere with the subject’s abilities to perform physical (e.g., hard and/or soft tissue trauma, including injuries to bone, muscle, ligament and tendons) or cognitive (e.g., color vision deficiency, intellectual disability) testing Taking medication that might interfere with the results of the study (e.g., corticosteroids, sleeping pills, anti-depressants, tranquillizers) Surgical intervention in the previous 3 months Regular consumption of tobacco, cannabis, or alcohol Special diet (e.g., vegetarian or vegan diet, food allergies) The presence any eating disorder according to the Dutch Eating Behavior Questionnaire (DEBQ)

**Table 2 nutrients-12-01885-t002:** Cognitive tests by neurocognitive domain and description.

Cognitive Test	Domain	Description
Stroop color–word test	Executive function	This test evaluates the ability to inhibit cognitive interference when the processing of a specific stimulus feature hinders the simultaneous processing of a second stimulus attribute. This test comprises three tasks (word reading, color naming, named color–word). First, participants are asked to name a series of color words. They are then asked to name the color of a bar (color task) of X’s (e.g., XXX in red, blue, or green ink). They are then asked to name the color of the ink rather than the word on colors printed in conflicting ink colors (e.g., the word “blue” in red ink).
Trail-making task (TMT) TMT-A TMT-B	Executive function	This test demands visual attention and motor speed. This task comprises two parts. In Part A, participants are asked to quickly draw lines on a page connecting 25 consecutive numbers. In Part B, participants are asked to connect the characters (numbers and letters) in an alternating order (i.e., 1, A, 2, B, 3, C, 4, D) as quickly as possible while still maintaining accuracy.
Barrage test	Attention/vigilance	This task evaluates visual-spatial ability and recognition. Participants must scan the required form in a sequence of forms. The duration of the test is 10 min. Its scores are awarded according to the speed and number of recognized symbols.
California verbal learning test (CVLT)	Memory	The CVLT will be administered according to the published method by Delis et al. (1987) [54]. The CVLT uses two different lists of words (A and B). Each list contains 16 words, four words each from four different categories presented pseudo-randomly. List A is presented five times, List B once immediately after the List A’s fifth presentation recall. For List A, the following measures will be made: The sum of correct items for each of the five trials and total correct for all five trials. For List B, the total correct is noted for the single presentation. The direct comparison of List A, first trial, with List B assesses proactive interference. Primacy and recency for recall are scored according to the CVLT standards: primacy (first four items); middle (interior eight items); and recency (last four items). Primary memory is the total recall of words for which the intra-trial retention interval is seven words or fewer. Secondary memory is the total recall of all words with intra-trial intervals greater than seven [55].
Multiple object tracking	Attention/executive function/working memory/processing speed	Multiple object tracking will be assessed using NeuroTracker (CogniSens Athletic Inc., Montreal, Canada) 3D multiple object-tracking device. For the task, participants wear 3D glasses and sit in a 3D simulator bay where they are asked to track object movements through space. The test consists of 20 trials in which speed of object motion is adjusted for subsequent trials based on prior scoring until a threshold is determined. The subject’s final score is calculated by averaging variable trial successes and failures depending on performance throughout the session.

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
