# Peer review of "Randomized Double-Blind Controlled Trial on the Effect of Proteins with Different Tryptophan/Large Neutral Amino Acid Ratios on Sleep in Adolescents: The PROTMORPHEUS Study"

_nutrients, 2020, doi:10.3390/nu12061885_

Round 1

Reviewer 1 Report

Thank you for allowing me to review the manuscript entitled "Randomized double-blind controlled trial on the effect of proteins with different tryptophan/large neutral amino acid ratios on sleep in adolescents: The PROTMORPHEUS study"

The manuscript is a study protocol that describes a double-blind RCT investigating on sleep disturbances in obese or elite rugby players of young age.

Introduction: well conceptualized. No further changes are needed

Materials and methods: reported methodology is meticolous. Only few concerns should be taken into account and resolved:

  • line 96: More information (including registration number) about local ethical committee should be added. 
  • line 360-361: cite the researches which were used as starting point for sample size calculation

Discussions: Robust, well written. Only a minor concern:

  • Together with the strenght of the protocol, authors should add what are the current limitations that will be applied.

Lastly, the manuscript should be checked by a native English speaker/writer.

Author Response

Response to reviewers:

            We thank the reviewers for their careful and thorough reading of this manuscript and for their thoughtful comments and constructive suggestions. This helped us to improve the quality of this manuscript. Our replies follow the reviewer’s comments in blue.

Reviewers' comments:

Reviewer 1

Comments and Suggestions for Authors

Thank you for allowing me to review the manuscript entitled "Randomized double-blind controlled trial on the effect of proteins with different tryptophan/large neutral amino acid ratios on sleep in adolescents: The PROTMORPHEUS study"

The manuscript is a study protocol that describes a double-blind RCT investigating on sleep disturbances in obese or elite rugby players of young age.

Introduction: well conceptualized. No further changes are needed

Materials and methods: reported methodology is meticolous. Only few concerns should be taken into account and resolved:

line 96: More information (including registration number) about local ethical committee should be added. 

Answer: We agree. Information about local ethical committee was added.

  “(Comité de protection des personnes (CPP) Sud Meditérrannée II, registration number: 2018 A23)”

line 360-361: cite the researches which were used as starting point for sample size calculation

Answer: We used the study realized by Ong, J.N.; Hackett, D.A.; Chow, C.-M. Sleep quality and duration following evening intake of alpha-lactalbumin: a pilot study#. Biol. Rhythm Res. 2017, 1–11.

We added it in the main text. Paragraph of statistical analysis have been rewritten to be clearer.

Discussions: Robust, well written. Only a minor concern:

Together with the strenght of the protocol, authors should add what are the current limitations that will be applied.

Answer: We agree. Besides the absence of laboratory polysomnography, we added the following limitations: “However, this study is also limited by a small sample size and do not include female subjects. Although it is interesting to look for gender effect, the control of menstrual cycle phase in adolescent girls would be complicated and incompatible with such a study design. Previous studies have shown that menstrual cycle phase impact both sleep, circadian rhythm, and food intake which could lead to biased assessments [64]. Finally, the effect of such an intervention in the medium and long term remain to be determined and would be interesting. Motivating the adolescents to follow this study protocol for a longer period is challenging and may severely penalized the feasibility of the study. However, previous studies already showed that dietary manipulation affect sleep significantly even over a shorter period of time than that of this study [65,66]”.

Lastly, the manuscript should be checked by a native English speaker/writer.

Answer: Manuscript was checked by a native English speaking.

Reviewer 2 Report

The paper “Randomized double-blind controlled trial on the 3 effect of proteins with different tryptophan/large 4 neutral amino acid ratios on sleep in adolescents: The 5 PROTMORPHEUS study” presents an interesting aspect of often debated relationships between sleep, metabolism, and nutrition. While the initial sample size could be bigger, all in all the paper has many merits that make the study well designed and justified. I really look forward to hearing results. However, I have some concerns that should first be addressed and discussed further.

ABSTRACT.

-If the journal protocol does not specifically state that this form of abstract is desired, please also indicate hypotheses and not just methods.

-Also, please check language – the word order is clumsy at times.  

INTRODUCTION.

-The introduction covers some central studies in both sleep research and nutrition. It could benefit from including studies which highlight external zeitgebers as influencers of circadian rhythms, and sleep quality.

-Please add a notion on the reasons why obesity and elite sports might be associated with poor sleep. Is this a question of nutrition, psychological aspects (such as anxiety or self-control), physiological (vagal, cardio-vascular), or metabolism? Some theoretical basics need to be covered in order for the hypotheses to be grounded and justified.

-Please add some recent literature on the topic, as some relevant papers have appeared (Nutrients. 2019 Apr 11;11(4):822. doi: 10.3390/nu11040822.)

-Also, please check language. Avoid statements such as “Some authors claim—“ as it is dismissive rather than objective (“One study reported—“ would be more neutral)  

METHODS.

-How are sleep problem diagnoses gathered? Self-reports?

-What are, in fact, classified as sleep disorders and is there a specific list of medication or illnesses that exclude participants? Allergies, oral surgery? Painkillers?

-And how do you define special diet? Are eating disorders considered at all?

-Please consider using Morningness-Eveningness Questionnaire for Children and Adolescents (MEQ-CA), it may be more suitable for this age group

-Please note that DLMO is difficult to measure reliably – you may wish to have controlled dark settings for diminishing the effects of light from outdoors and different devices

-This age group is notorious for late sleep onset. How are you going to motivate the adolescents to follow the set schedules for such a long time? Sleep rhythms may also differ significantly before the wash-out period; 3 days will not be enough for adjustment.

DISCUSSION.

-Please mention that also the obese group may have pressure in their lives, there is often comorbidity regarding anxiety etc. 

-Please discuss the potentially distorting effect of latent factors leading to obesity

Author Response

Response to reviewers:

            We thank the reviewers for their careful and thorough reading of this manuscript and for their thoughtful comments and constructive suggestions. This helped us to improve the quality of this manuscript. Our replies follow the reviewer’s comments in blue.

Reviewers' comments:

Reviewer 2

The paper “Randomized double-blind controlled trial on the 3 effect of proteins with different tryptophan/large 4 neutral amino acid ratios on sleep in adolescents: The “PROTMORPHEUS study” presents an interesting aspect of often debated relationships between sleep, metabolism, and nutrition. While the initial sample size could be bigger, all in all the paper has many merits that make the study well designed and justified. I really look forward to hearing results. However, I have some concerns that should first be addressed and discussed further.

ABSTRACT.

-If the journal protocol does not specifically state that this form of abstract is desired, please also indicate hypotheses and not just methods.

Answer: Hypotheses were added as follow:” Ingested Trp is involved in the regulation of the sleep/wake cycle and improvement of sleep quality. Since Trp transport through the blood-brain barrier is competing with LNAAs, protein with higher Trp/LNAAs were expected to increase sleep efficiency.”.

-Also, please check language – the word order is clumsy at times

Answer: Manuscript was checked by a native English speaking.

INTRODUCTION.

-The introduction covers some central studies in both sleep research and nutrition. It could benefit from including studies which highlight external zeitgebers as influencers of circadian rhythms, and sleep quality.

Answer: We adapted a paragraph in the introduction where we added studies focusing on nutrition as external zeitgeber and how it could affect both circadian clock and sleep.

“To date, few non-pharmacological approaches have been implemented to prevent sleep disturbances in these populations. A wide range of papers emphasize that external Zeitgeber such as nutrients and physical activity play a part in the regulation of the internal clock and sleep quality [15,16]. Nutrients reprogram peripheral clocks such as the liver or the gut clock by exerting an effect on the transcriptional and translational regulation of molecular clocks (e.g. CLOCK/BMAL) which generates a considerable reorganization of homeostasis and specific metabolic pathways [17]. Moreover, we have recently noted a growing interest on the effect of dietary intake on sleep and several reviews have been published over the last decade [18–22]”.

-Please add a notion on the reasons why obesity and elite sports might be associated with poor sleep. Is this a question of nutrition, psychological aspects (such as anxiety or self-control), physiological (vagal, cardio-vascular), or metabolism? Some theoretical basics need to be covered in order for the hypotheses to be grounded and justified.

Answer: we added a paragraph in the introduction to justify why obesity and elite sports might be associated with poor sleep.

“Besides endogenous factors linked to maturational changes during adolescence which push for a delay in the timing of sleep and increase the risk of sleep disturbances [1]. Other external factors specific for each population (obese or athlete) exacerbate poor sleep outcomes. In adolescents with obesity the involvement of a complex interaction between sleep, obesity, metabolic disruption and low grade inflammation where advanced [12,13]. On the other hand, the increased time demands by elite sports pressures including high training demands, evening intense exercise, competition, travel, strict schedules contribute to excessive stress, anxiety, physiological excitement (e.g. increased heart rate at bedtime), pain, and muscle soreness which impair sleep in in young athletes [7,14]”.

-Please add some recent literature on the topic, as some relevant papers have appeared (Nutrients. 2019 Apr 11;11(4):822. doi: 10.3390/nu11040822.)

Answer: We agree. Two sentences citing further studies were added.

“Previous studies on tryptophan depletion have shown that decreasing circulating tryptophan impairs sleep (e.g. increased arousals, delayed REM sleep) [32,33]. Moreover, it was suggested that small amount of dietary Trp intake could improve sleep efficiency in insomniac subjects [34]. The pilot study by Ong et al. (2017) also reported favorable effect of Trp rich protein on accelerometry measured sleep among young adults without sleep complaints [35]”.

-Also, please check language. Avoid statements such as “Some authors claim—“ as it is dismissive rather than objective (“One study reported—“ would be more neutral)  

Answer: We changed in accordance to reviewer’s recommendation.

METHODS.

-How are sleep problem diagnoses gathered? Self-reports?

Answer: Although, no laboratory polysomnography will be set to explore sleep before the study. During the inclusion, an examination will be carried out by the investigating physician in order to affirm or not the presence of symptoms of major sleep disorders (e.g. narcolepsy), (2) incompatible medical history (i.e. major depressive disorder, children obstructive sleep apnea with Apnea-Hypopnea Index>5) and related treatments (e.g. antidepressants, anxiolytics, corticosteroids) that could interfere with sleep outcomes. An interview will be held with the potential participants as well as their legal guardians (parents) during which questionnaires will be also completed. As stated in table 1, High risk of Obstructive Sleep Apnea (i.e. Berlin questionnaire with 2 or more categories where the score is positive) or any history of sleep disorders (e.g. narcolepsy, children obstructive sleep apnea with Apnea-Hypopnea Index>5) will be considered as an exclusion criterion.

-What are, in fact, classified as sleep disorders and is there a specific list of medication or illnesses that exclude participants? Allergies, oral surgery? Painkillers?

Answer: Roughly speaking, besides obesity (in the adolescent with obesity group) participants should be in good health. As already stated in table 1, high risk of Obstructive Sleep Apnea (i.e. Berlin questionnaire with 2 or more categories where the score is positive), already diagnosed or suspicion of major sleep disorders (e.g. narcolepsy, children obstructive sleep apnea with Apnea-Hypopnea Index>5) were considered as exclusion criteria. In order to be more specific a more extensive list was added including (Restless Leg Syndrome, REM Sleep Behavior Disorder, Bruxism) for sleep disorders. Moreover, we added specific examples regarding Medical history not consistent with the study as participants were screened for depression using the Kutcher Adolescent Depression Scale (KADS) (LeBlanc et al., 2002) (also added in the main text), the list also includes common pediatric mental disorders that could also interfere with sleep outcomes (Attention-Deficit/Hyperactivity Disorder, Post-traumatic Stress Disorder) or concussion particularly in the case of junior rugby players. We also specified some chronic illness or injury that might interfere with the subject’s abilities to perform physical (e.g. hard and/or soft tissue trauma, including injuries to bone, muscle, ligament and tendons), or cognitive (e.g. color vision deficiency, intellectual disability) testing. For allergies, please see the next answer.

-And how do you define special diet? Are eating disorders considered at all?

Answer: Special diet include a vegetarian or vegan diet as well as food allergies since the meal menu was planned in advance. This has been specified in table 1.

Effectively, given that the evaluation day includes Ad libitum 24 h energy intake, participants were screened for eating disorder according to the Dutch Eating Behavior Questionnaire (DEBQ) (Van Strien et al., 1986). This has been added to the text (line 116) and table1.

-Please consider using Morningness-Eveningness Questionnaire for Children and Adolescents (MEQ-CA), it may be more suitable for this age group

Answer: We agree and amended in accordance to reviewer’s recommendation (line 173 and line 237).

-Please note that DLMO is difficult to measure reliably – you may wish to have controlled dark settings for diminishing the effects of light from outdoors and different devices

Answer: Effectively DLMO will be captured in Dim light conditions (this was added to the text). After dinner, the participants will enter the light-controlled study room <20 lux in any direction of gaze (in the same institution where they live), and hourly saliva samples will be taken. Although challenging, while the participants provide saliva samples, experienced sleep technicians will equip them with EEG monitors. Then, they will be accompanied to their rooms for sleep avoiding exposure to any bright lights.

Given the potential effect of electronic media (smartphone, laptop...) on sleep and circadian rhythm, their evening use will be prohibited during the study. This sentence was added to the main text (line 182).

-This age group is notorious for late sleep onset. How are you going to motivate the adolescents to follow the set schedules for such a long time? Sleep rhythms may also differ significantly before the wash-out period; 3 days will not be enough for adjustment.

Answer: Strict sleep schedules are already set up in the institutions where the participants live from (from 10:30 pm to 7:00 am). For the study, bedtime will be only advanced by 30 min.

Each group will undergo the advanced sleep schedule one week prior to the experimental sessions launch, time in bed (TIB) will be fixed (from 10:00 pm to 7:00 am), and sleep/wake cycle will be continuously monitored using accelerometry, in order to make sure that this advance would be reached.

DISCUSSION.

-Please mention that also the obese group may have pressure in their lives, there is often comorbidity regarding anxiety etc. 

Answer: We agree. Studies underlined that adolescents with obesity were particularly prone to to several psychopathologic conditions, including anxiety and chronic stress which affect the hypothalamic–pituitary–adrenal axis and increase the arousal/sympathetic nervous systems activities. This sentence was added to the main text (line 448). We also added the reference.

-Please discuss the potentially distorting effect of latent factors leading to obesity

Answer:  We agree. A sentence was added to the last paragraph of the discussion as follow:

“Latent factors related to both obesity and/or sleep may affect outcomes of studies investigating the effect of dietary intervention on sleep. For instance, we already showed in previous works that physical activity could affect sleep in adolescents with obesity [63]. Therefore, we underline that these factors should be systematically quantified and added as confounders”.